# Genetic Identity Based on Whole-Genome SNP Array Data of Weedy Rice in Nagano, Japan

**Wenjing Bi [1,2], Jian Sun [1], Jun Hosoi [3], Masaharu Aoki [3], Nagao Sakai [3], Tomio Itani [4], Zhengjin Xu [1,]\* and Tohru Tominaga [2,]\*** 

1   Rice Research Institute, Shenyang Agricultural University, Shenyang 110866, China
2   Graduate School of Agriculture, Kyoto University, Kyoto 6068502, Japan
3   Experimental Station of Nagano Prefecture, Suzaka, Nagano 3820051, Japan
4   Faculty of Agriculture, Ryukoku University, Otsu, Shiga 520-2194, Japan
\*   Correspondence: xuzhengjin2011@hotmail.com (Z.X.); tominaga@kais.kyoto-u.ac.jp (T.T.)

**Abstract:** The rice production system in Japan is changing due to the aging of rice farmers, shortage of labor, mechanization, and increase of large-scale rice fields and increased application of direct sowing, all of which contribute to the lack of sophisticated weed management practices. Because the changing production system has not improved and likely worsened weed management practices, weedy rice (*Oryza sativa* L.) has become a serious threat to rice production in Japan. We analyzed weedy rice accessions from Nagano, the central part of Japan, and other rice accessions to elucidate the origin of this weedy rice using a whole-genome single nucleotide polymorphism (SNP) array. For developing sustainable weedy rice management practices, the elucidation of the origin of re-emerging weedy rice is crucial. Population genetic analysis indicated that weedy rice in Nagano was phylogenetically independent from the local red rice cultivar with low genetic diversity. Past and recently re-emerging weedy rice ecotypes in Nagano shared a similar genetic background which implies the evolution of weedy rice by severe natural selection. Window-based $F_{ST}$ and selection sweep analysis revealed the divergence of some of the known key domestication-related genes, such as shattering gene *sh4* and *qsh1*, of weedy rice from domesticated rice cultivars.

**Keywords:** de-domestication; local adaptation; origin; red rice; weedy rice; whole-genome SNP array

## 1. Introduction

In crop production, weeds are typically controlled by the application of herbicides. In modern agriculture, selective herbicides are used in crop production to more effectively reduce weed populations. Herbicide selectivity between crops and weeds is conferred by the differences in growth stage, growth characteristics, and physiological properties between the crop and weed. However, this weed-control strategy is undermined when weeds are too closely related to the crop. For example, the origin of weedy rice is wild rice, volunteer cultivars, and/or hybrids between wild rice and cultivar [1,2]. Weedy rice belongs to the same species as cultivated rice and, consequently, is problematic in almost all rice-producing countries. Weedy rice severely shatters seeds, but there are few ecological and physiological differences between the two types of rice. Hence, a major difficulty to control weedy rice is the scarcity of effective selective herbicides against weedy rice that will not harm cultivated rice.

Weedy red rice was reported in Japanese rice paddy fields in the 1970s and was threatening rice production [3]. At that time, weedy red rice was easily distinguishable from cultivated rice by its red color of hulls, and through intensive hand-weeding, weedy red rice was eliminated from rice paddy fields. However, a potentially new weedy rice with faded red hulls and long awns has recently re-emerged in some parts of Japan. It is not clear whether the recently re-emerging weedy rice

population has an independent origin from or is closely related to the early populations of weedy rice found in the 1970s.

The re-emergence of weedy rice is likely closely related to the changes of the rice production system in Japan, a system which is lacking sophisticated, careful weed management strategies due to the aging of rice farmers, shortage of labor, and mechanization and increase of large-scale rice fields. One possible cause of the re-emergence is the shift from hand-weeding to the use of herbicides, especially the use of one-shot herbicide. Additionally, to distinguish the new weedy rice with faded red hulls from cultivated rice is not easy, so the new weedy rice sometimes could escape from hand-weeding. A second change is the spread of application of the direct sowing technique in rice cultivation. In Japan, the method of transplanting is most commonly used while direct sowing only accounted for less than 0.5% of the cultivated areas of rice. However, direct sowing is increasingly practiced in some parts of Japan because of its lower cost in rice production. A third change is that rice is now often cultivated for livestock. Due to a decrease in the human consumption of rice, some paddy fields are used to produce whole-crop silage of rice plants for livestock feed [4]. In rice paddy fields, cultivars with high biomass production are cultivated and these cultivars are sometimes hybrids between *japonica* and *indica* rice, which can relatively easily shatter like weedy rice does.

Red rice is another potentially closely-related variety to weedy rice. It has been traditionally cultivated for religious ceremonies in shrines and is sometimes cultivated for sake or Japanese rice wine. Recently, rice growing in paddy fields has been used as a new medium of art; red rice is arranged in various ways to use the red color as in paintings and the field as the canvas. Similar to weedy rice, some red rice cultivars also relatively easily shatter.

The origin of the recently re-emerging weedy rice is unclear and we suspect potential genetic relationships between the re-emerging weedy rice and the red rice cultivar or rice grown for livestock feed. Elucidating the origin of the re-emerging weedy rice is crucial for developing sustainable weedy rice management practices and understanding the evolution of weedy rice is of interest to both rice producers and researchers. Therefore, we investigated the possible origins of the re-emerging weedy rice using data obtained from a whole-genome single nucleotide polymorphism (SNP) array.

## 2. Materials and Methods

### 2.1. Plant Materials

A total of 44 weedy rice accessions were collected for the present study; 37 were sampled from Nagano, Japan and henceforth, collectively named as WRN (weedy rice accessions sampled from Nagano, Japan). Of the WRN accessions, 32 were re-emerged weedy rice lines sampled in 2015, and five were weedy red rice lines sampled in the 1970s. In addition, we also sampled 28 accessions of cultivated *japonica* rice (CJR) and 16 accessions of traditional red rice (TRR). Four *indica* and four *aus* accessions were used as two separate control groups. All of the rice samples were cultivated in the experimental field of Kyoto University, Kyoto, Japan.

### 2.2. Information for SNP Genotyping Array

To dissect the genetic identity of Nagano weedy rice, we genotyped 44 WRN, 28 CJR, 16 TRR, 4 *aus*, and 4 *indica* by using a 50K SNP array that combined two sets of commercial rice SNP chips, a 44K chip, and RICE6K [5,6]. The 44K chip was generated from two data sources, the OryzaSNP project and BAC clone Sanger sequencing of wild species from the Oryza Map Alignment Project (OMAP), which was adequately powered to further our dissection of the population structure of *O. sativa* [7,8]. This 44K chip included 44,100 well-distributed SNPs with ~1 SNP per 10 kb across the 12 chromosomes of rice. RICE6K was developed based on re-sequencing data of more than 500 rice landraces. The RICE6K chip included 5636 representative SNPs that were selected from more than four million SNPs, and the chip was suitable for rice germplasm fingerprinting, genotyping bulked segregating pools, seed authenticity check and genetic background selection. Therefore, the combined

chip with 49,656 SNPs (named as 50K SNP) used in this study should be sufficient to dissect the genetic identity of Nagano weedy rice.

## 2.3. DNA Isolation

DNA was extracted from young leaves of each rice plant using the Qiagen plant DNeasy protocol. Then the genomic DNA was quantified by the NanoDrop ND-2000 (Thermo Scientific, Tokyo, Japan), and the DNA integrity was assessed using agarose gel electrophoresis.

## 2.4. Array Hybridization

The genotyping process included the following seven steps: sample labeling and then DNA amplification, fragmentation, precipitation, resuspension, microarray hybridization, and washing. Hybridization Master Mix was added after gDNA resuspension following a quality control (QC) of samples. Samples that passed the QC were hybridized, washed, and then scanned on the GeneTitan MC Instrument (Affymetrix, Tokyo, Japan). All the steps were performed according to the protocols provided by Affymetrix.

## 2.5. SNP Genotype Calling

Raw data were generated from the GeneTitan system. Then we loaded them to the Axiom Analysis Suite software to perform clustering and genotype calling. A final custom report was created from the Axiom Analysis Suite for the downstream analyses; we obtained the SNP set named 50K SNP, which included 50,281 SNPs with a minor allele frequency (MAF) > 0.05 and calling rate > 0.92.

## 2.6. Population Structure and Genetic Diversity

A phylogenetic tree was generated with MEGA 7.0 from a pairwise-distance matrix using the neighbor-joining method to reveal the genetic relationships between weedy rice and control groups [9]. Bootstrap values correspond to 1,000 replications. Principal component analysis (PCA) based on 60K SNP was carried out with the Biomarker biocloud platform (http://www.biocloud.net/). Diversity estimates ($\pi$ and $\theta$) were obtained using the TASSEL 5.0 software [10].

## 2.7. Genome Divergence, Selection Sweeps, and Gene Flow Analysis

The window-based $F_{ST}$ and nucleotide diversity ($\pi$) among WRN, CJR, and TRR were calculated based on 50K SNPs by using the vcftools software [11]. Then we drew a histogram and line chart to define the regions of genomic divergence and selection, respectively. The 95th percentile of all sliding-window $F_{ST}$ values were identified as outliers throughout the genome. The occurrence of gene flow was determined by the *D*-value and D-statistic value significantly deviating from zero [12].

## 3. Results

### 3.1. SNP Genotyping and Population Structure

We performed clustering of the raw data using the Axiom Analysis Suite software for SNP genotype calling of all 96 rice samples. All samples passed quality control with values of Dish QC > 0.82 and calling rate >0.92, resulting in a total population of 50,281 SNPs obtained for the following analysis.

To reveal the genetic relationships between sampled weedy rice and other rice groups, we constructed a bootstrapped neighbor-joining tree inferred from 1000 replicates that were based on the 50K SNP chip. The neighbor-joining tree showed that the 96 rice samples exhibited significant phylogenic divergences corresponding to rice subspecies differentiation. All 32 modern WRNs (weedy rice accessions sampled from Nagano, Japan) independently clustered together with four early WRNs in the *japonica* branch except for one *indica* WRN. However, non-Nagano weedy rice did not cluster in this group, which implies that the genetic identity of weedy rice samples may have been

geographically isolated to the Nagano area. Whereas TRR (traditional red rice) distributed widely in this neighbor-joining tree, interestingly, only one TRR grouped together with the WRNs that originated from China (Figure 1). In addition, WRN accessions were further separated into sub-groups based on the radiation type of the neighbor-joining tree (Figure 2).

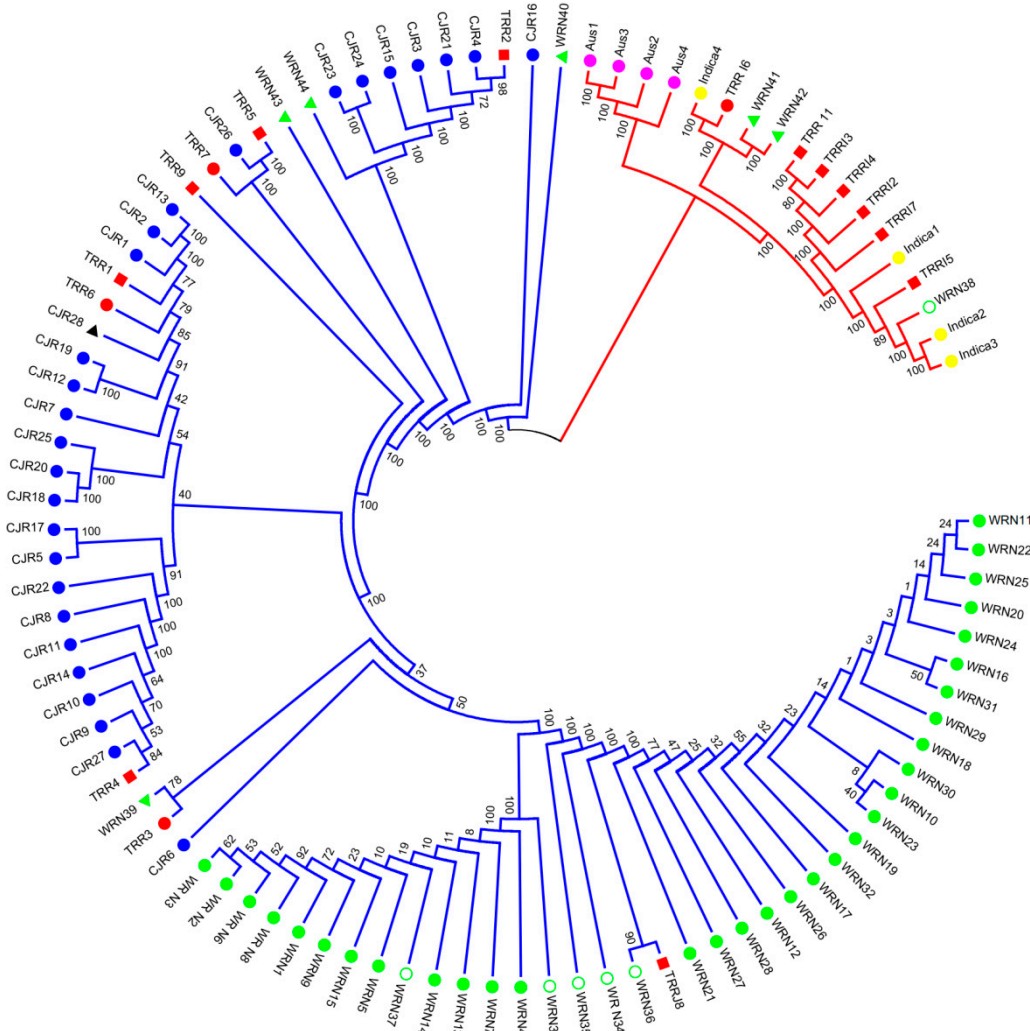

**Figure 1.** Neighbor-joining tree built from 50K SNPs (single nucleotide polymorphisms). Numbers at the branch nodes represent bootstrap values. Blue- and red-colored branches indicate the *japonica* and *indica* groups, respectively. Solid and open green circles indicate WRN (weedy rice accessions sampled from Nagano, Japan) samples of the modern and early races, respectively. Non-Nagano weedy rice accessions are indicated by green triangles. Blue circles indicate CJR (cultivated japonica rice). Red circles and red squares indicate Japanese TRR (traditional red rice) and Chinese TRR, respectively. *Indica* and *aus* are indicated by yellow and pink circles, respectively. The black triangle indicates the *japonica* rice Koshihikari.

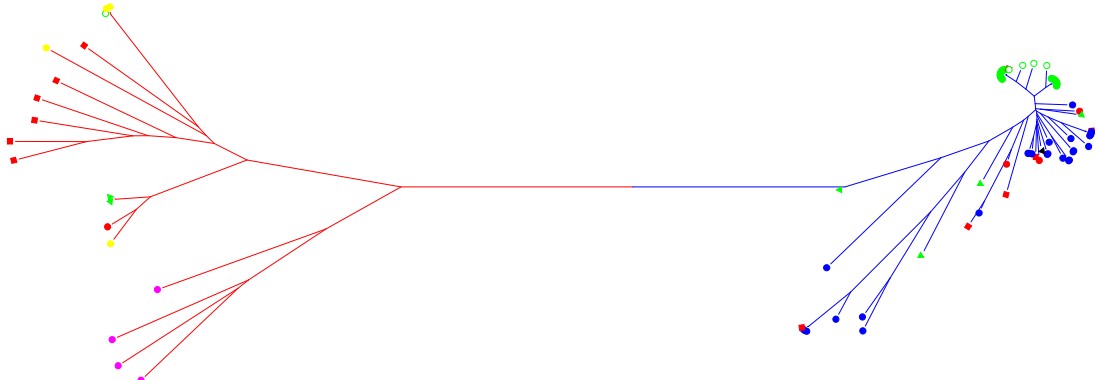

**Figure 2.** Radiation type of neighbor-joining tree constructed from 50K SNPs. Symbols represent the same samples as described in Figure 1. Note the sub-groups of WRN (open and closed green circles).

To confirm that WRN belongs to a group of *japonica* rice based on phylogenic analysis, we conducted a PCA of the *japonica* group to further dissect the genetic relationships of WRN with TRR and CJR (cultivated japonica rice). In the PCA as shown in Figure 3, PC1 (the first principal component) separated TRR and CJR into two groups and explained 52.52% of the genetic variance, whereas PC1 and PC2 (36.15% of the variance) together identified the WRN samples as a distinct gene pool. The variance of the separation within WRN as explained by PC3 was 16.33%.

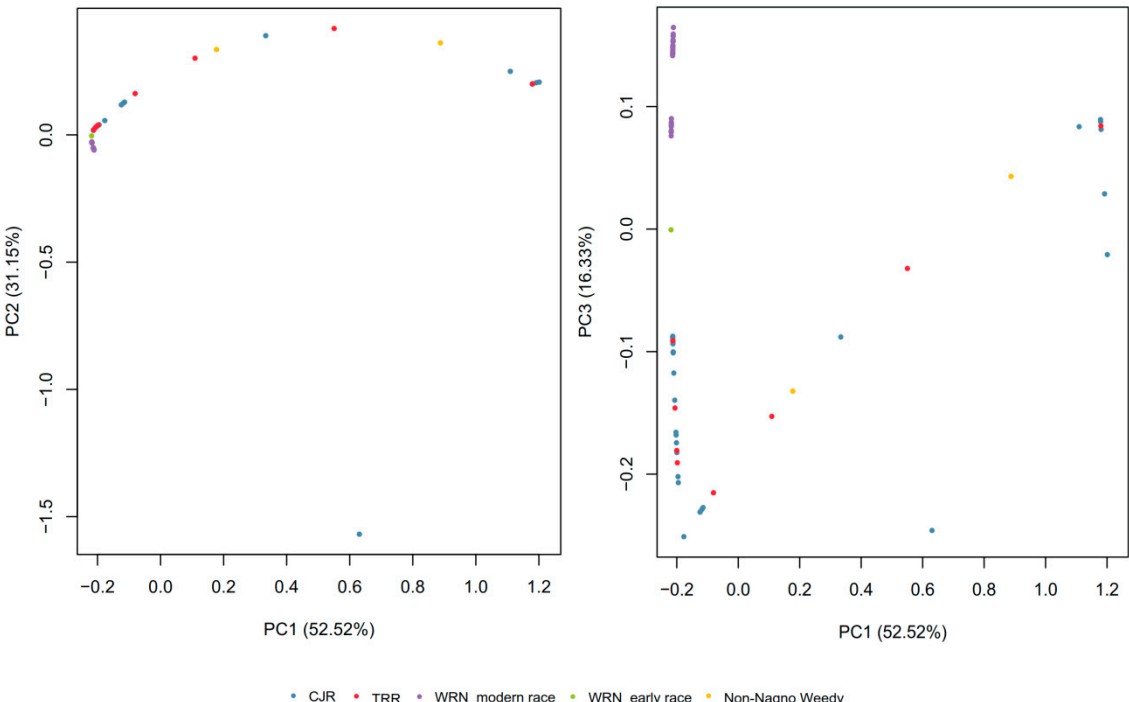

**Figure 3.** Principal component analysis plots of the first three components of the *japonica* group. The percentage of genetic variance explained by each principal component are included in the axis labels.

*3.2. Gene Diversity and Gene Flow*

The diversity estimate of $\pi$ was used to assess the level of genetic polymorphism, which was determined for the WRN, CJR, and TRR populations. The $\pi$ values of CJR and TRR were 0.123 and 0.116, respectively. Levels of polymorphism in WRN were predictably lower, $\pi = 0.062$, than that of the two control groups. Gene diversity was reduced approximately by half of the total WRN population

($\pi$ = 0.033) when only considering the modern WRN. Then we used the *D*-statistic value for further exploration of whether gene flow with CJR and TRR may have contributed to the emergence of WRN (Table 1). The *D*-statistic value (0.0001) did not support that gene flow occurred because it did not significantly deviate from 0. The above analysis implies that WRN is a unique group with conservative diversity due to the regional adaptability rather than belonging to certain branch of TRR of Japan.

**Table 1.** Gene flow determination among different groups.

|  | Pop1 | Pop2 | Pop3 | 0pop | *D* | *Z* | BABA | ABBA |
|---|---|---|---|---|---|---|---|---|
| Result | CJR | TRR | WRN | Outgroup | 0.0001 | 0.023 | 2845 | 2845 |

### 3.3. Genome Divergence and Selection Sweeps

Current genetic evidence is insufficient to determine the ancestral relationships among the three groups, WRN, CJR, and TRR. For this reason, we selected the window-based $F_{ST}$ using the window-based ratio of nucleotide diversity ($\pi$) to detect divergent genomic regions. We used 100 kb windows for two comparisons, WRN vs. CJR and WRN vs. TRR, and determined the genome-wide distribution of significant divergent regions with a threshold set to the top 1% of the window $F_{ST}$ (Figure 4A,B). Some of the known key domestication-related genes are located in the common diverged regions deduced by the two comparisons, such as shattering gene *sh4* and *qsh1* [13–15]. Some regions of divergence only occurred in one of the two comparisons, for example, a significant divergence signal in *wx* (gene related to amylose content) regions was only detected in the comparison of WRN to CJR [16]. Isolation of the agronomic- and adaptation-related candidate genes in these divergence regions will deepen our understanding of the genetic identity of weedy rice found in Nagano.

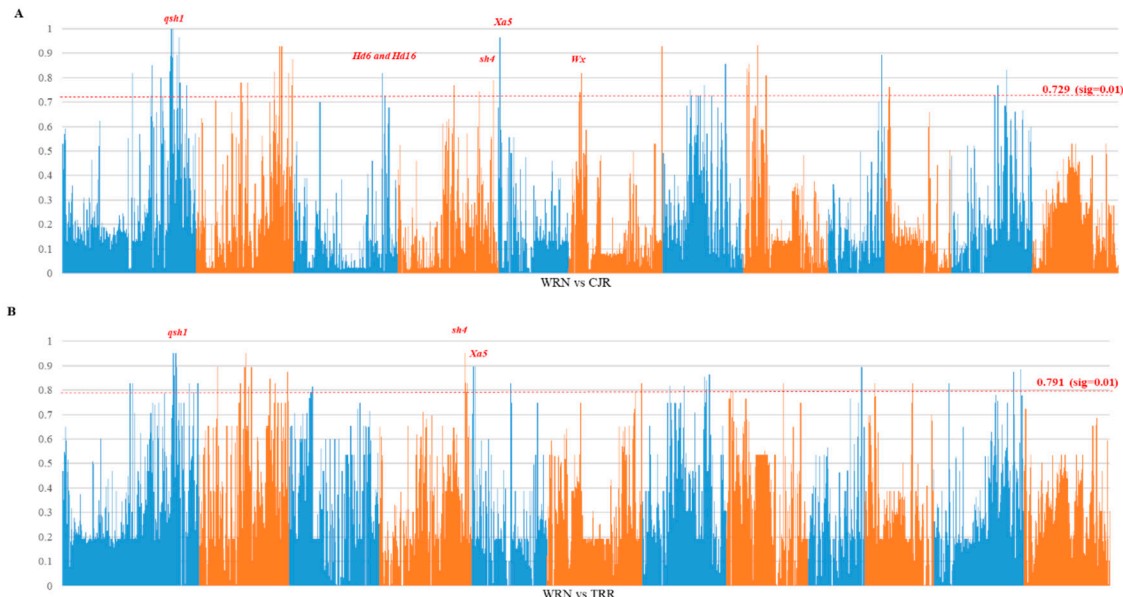

**Figure 4.** Genomic divergence between WRN and each of two control groups. Histogram based on a 100 kb sliding-window $F_{ST}$ comparisons between (**A**) WRN and CJR and (**B**) WRN and TRR. The dotted red line indicates the cutoff for the top 1% of windows.

## 4. Discussion

### 4.1. Origin of Weedy Rice in Nagano

Ascertaining the origin of weedy rice has been controversial. Several possible paths of its origin in different rice-producing regions of the world have been reported, such as (i) introgression from

wild rice to cultivated rice; (ii) direct spread of wild rice to cultivated fields with adaptation feature; and (iii) introduction from exotic germplasm into the local paddy germplasm [17–20]. The latest mainstream opinion, based on NGS (next-generation sequencing) technology, is that 'de-domestication', the conversion of an organism from the domestic state into a wild-like variety, gave rise to both weedy rice found in the U.S. and China [1,2,21,22].

In Japan, no wild relatives or ancestors of cultivated rice were intentionally grown, thus there was likely no opportunities for wild rice to have directly spread into cultivated fields and outcross with local cultivars. Though there are some *japonica* landraces possessing pre-domestication traits and genes which are called 'heritage landraces' [23,24]. In the present study, the genetic structure showed that weedy rice of Nagano shared a similar genetic background with these 'heritage landraces'. We speculate that the origin of weedy rice in Nagano was an introduction from an exotic germplasm, and it re-emerged due to the increase in fitness with the modern changes in rice cultivation practices [4].

Our genome-scale study based on SNP array data indicates that weedy rice in Nagano is phylogenetically independent of the local cultivar with low genetic diversity. The 1970s and modern accessions of weedy rice in Nagano shared a similar genetic background which implies that the potential evolution of Nagano weedy rice was shaped by severe selection. On the other hand, with the occurrence of climate change, extreme weather has gradually appeared in the rice growing season in recent years leading to, for example, fertility disorders in cultivated rice often caused by extreme high temperature and low temperature at the booting and heading stages. However, the growth period of weedy rice was inconsistent with cultivated rice; this enhanced the competitiveness of weedy rice and increased its population fitness, thus accelerating its evolution.

### 4.2. Strategies for Weedy Rice Control

Transplanting rather than direct seeding of rice can suppress weedy rice infestations to a certain degree. Despite this knowledge, the direct seeding method has been widespread, which has promoted the rapid spread of weedy rice with the unfortunate consequence of these invasions being difficult to control effectively [25]. The weedy red rice in Japan was easy to remove by hand and achieve eradication from rice paddy fields due to its discriminating red color. However, the recently re-emerging populations of weedy rice in some parts of Japan display a less distinguishable color type from that of cultivated rice, causing tremendous difficulty to identify and remove it by hand-weeding [26]. Methods that are helpful to try to control infestations of weedy rice in cultivated fields are crop rotation, minimum tillage, hand removal of panicles at the heading stage, seed bed preparation with moldboard plowing, and water management [27,28]. In a transplanted system, herbicide application prior to rice transplanting is also effective [29]. The seed survival of Nagano weedy rice in soil is three years [30]. Therefore, at least three years of rigorous removal is crucial to achieve eradication of weedy rice.

### 5. Conclusions

In this study, we clarified that Nagano weedy rice phylogenetically independent from the local red rice cultivar with low genetic diversity and presented conservative evolution due to the severe natural selection and negative artificial selection. Domestication-related genes may contribute to the population differentiation between weedy rice and cultivated rice based on selection sweep analysis.

**Author Contributions:** Conceptualization, T.T.; Data curation, W.B. and J.S.; Formal analysis, W.B. and J.S.; Funding acquisition, W.B.; Investigation, W.B., J.S. and T.T.; Methodology, J.S., Z.X. and T.T.; Project administration, Z.X. and T.T.; Resources, J.H., M.A., N.S., T.I. and T.T.; Supervision, Z.X. and T.T.; Validation, J.S., Z.X. and T.T.; Visualization, W.B. and J.S.; Writing—original draft, W.B., J.S. and T.T.; Writing—Review & Editing, T.T. and J.S.

**Funding:** This research received no external funding. The APC was funded by Shenyang Agricultural University.

**Acknowledgments:** W.B. was supported by the China Scholarship Council.

**Conflicts of Interest:** The authors declare that they have no conflict of interests.

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
