# Peer review of "Genetic Identity Based on Whole-Genome SNP Array Data of Weedy Rice in Nagano, Japan"

_agronomy, doi:10.3390/agronomy9080472_

Round 1
Reviewer 1 Report
The work of Wenjing and collaborators addresses an important topic for rice production, not only in Japan, but globally, as it refers to evolutionary changes that might be underexploited in other rice producing regions. Thus, this work call up to the relevance of similar studies, particularly taking into account the permanent environmental changes and pressure.
In my opinion, the MS is well written and structured, with a clear presentation of the results and the associated relevance and novelty.
I would suggest the following minor changes:
L 18-19: “For developing sustainable weedy rice management practices, to elucidate the origin of the re-emerging weedy rice is very important”, replace by “…..the elucidation of the origin of re-emerging weedy rice is crucial”
L 22: “…rice strains……” replace by “…rice ecotypes…”
L36: “.. is the same species...” replace by “…belongs to the same species….”
L 38: “…in control of weedy...” replace by “…to control weedy…”
L 95: “….is used to assess...” replace by “…was used…”; “….which we determined…..” replace by “…which were determined….”
L 103: “….rather than belong to certain branch…” replace by “….rather than belonging to a certain….”
L 129: “….and are....” replace by “….which are....”
Line 131: “....however, this study provides the sole evidence.” Could this sentence be clarified or omitted?
Line 133: “in a dormant state, and it re-emerged due to the modern changes in rice cultivation practices…” Could few supporting references of this hypothesis be included?
Line 134: “…in Nagano are...” replace by “…in Nagano is…”
Line 136: “….which implies the potential....” replace by “…which implies that the potential….”
Line 140: “…there has been an increase in use of the direct seeding method which has promoted…” replace by “….the widespread of the direct seeding method has promoted…..”
In general: Is it possible to address in the discussion the evolutionary aspects of weedy rice within the context of climate changes?
Author Response
Dear Reviewer 1,
We thank you very much for your valuable and constructive comments. We have addressed all of the issues raised by you and the reviewer 2, and yellow-highlighted the revised parts.
We have replaced words and sentences according to your comments.
Line 131: We have omitted the sentence "..... however, this study provides the sole evidence" according to your comments.
Line 133: We have omitted the sentence "that has survived in paddy soils in a dormant states" and added a reference.
Line 142-147: We have added the discussion on the context of climate change.

Reviewer 2 Report
Agronomy 574311
The authors have conducted a study to ascertain the origin of weedy rice that is currently found in Japan. They have conducted an adequate experiment to get to this issue.
The Introduction is a little scattered n its presentation of the issues.
The current weedy rice, like its predecessors, is a problem. The previous iteration of weedy rice was red in colour, and as such was able to be hand removed from the fields. There were no herbicides available then to control (similar to today’s issue), therefore the problem, as stated in lines 38-40, is not selectivity of herbicides, but a later described is the change from intensive hand weeding to extensive cultural practices.
Is it possible that the presence of this ‘new’ weedy rice is that due to its colour, it escaped hand weeding and has persisted in its presence until it has been recently noticed once populations reached high enough levels? This could explain its closer similarity to TRR (Section 2.3) as compared to NJP.
Specifics:
Lines 48 -60: what are sophisticated weed management strategies? It appears to be in large part referring t hand-weeding, which would to be considered sophisticated. Would hand-weeding successful in controlling the current weedy rice?
Section 2 - Results:
Define the population WRN, TRR and CJR, this is only defined in later sections (Section 4) and causes the reader to have to flip back and forth across pages and takes away from the flow of the paper when being read.
Section 3 - Discussion:
There is not a concluding paragraph for this paper.
Lines 149-150: Seed vigour or is seed persistence in the soil what is meant here?
Section 4 - Materials and Methods:
Section 4.5 SNP genotype calling. What was the missing data cut-off level? Minor gene occurrence threshold?
Author Response
Dear Reviewer 2,
We thank you very much for your valuable and constructive comments. We have addressed all of the issues raised by you and reviewer 1, and yellow-highlighted the revised parts.
Line 49, 52-54: We have added "careful" and the sentence stated on the new weedy rice with faded red hulls according to your suggestions.
We have defined WRN, TRR and CJR.
We have added the conclusion in Line 164-168.
Line160: We have replaced "vigour" to "survival".
Section 4.5: We have added "which included 50,281 SNPs with a minor allele frequency (MAF)>0.05 and calling rate>0.92.
